# Using DWS Optical Readout to Improve the Sensitivity of Torsion Pendulum

**DOI:** 10.3390/s23198087

**Published:** 2023-09-26

**Authors:** Shaoxin Wang, Heshan Liu, Lei Dai, Ziren Luo, Peng Xu, Pan Li, Ruihong Gao, Dayu Li, Keqi Qi

**Affiliations:** 1Center for Gravitational Wave Experiment, National Microgravity Laboratory, Institute of Mechanics, Chinese Academy of Sciences (CAS), Beijing 100190, China; wangshaoxin@imech.ac.cn (S.W.); luoziren@imech.ac.cn (Z.L.);; 2Taiji Laboratory for Gravitational Wave Universe (Beijing/Hangzhou), University of Chinese Academy of Sciences (UCAS), Beijing 100049, China; 3Changchun Institute of Optics, Fine Mechanics and Physics, Chinese Academy of Sciences (CAS), Changchun 130033, China; 4Hangzhou Institute for Advanced Study, University of Chinese Academy of Sciences (UCAS), Hangzhou 310024, China

**Keywords:** torsion pendulum, sensitivity, optical readout, DWS

## Abstract

In space gravitational wave detection missions, a drag-free system is used to keep the test mass (TM) free-falling in an ultralow-noise environment. Ground verification experiments should be carried out to clarify the shielding and compensating capabilities of the system for multiple stray force noises. A hybrid apparatus was designed and analyzed based on the traditional torsion pendulum, and a technique for enhancing the sensitivity of the torsion pendulum system by employing the differential wavefront sensing (DWS) optical readout was proposed. The readout resolution experiment was then carried out on an optical bench that was designed and established. The results indicate that the angular resolution of the DWS signal in optical readout mode can reach the level of 10 nrad/Hz^1/2^ over the full measurement band. Compared with the autocollimator, the sensitivity of the torsional pendulum is noticeably improved, and the background noise is expected to reach 4.5 × 10^−15^ Nm/Hz^1/2^@10 mHz. This method could also be applied to future upgrades of similar systems.

## 1. Introduction

The observation of the GW150914 double black hole merging event by the ground-based gravitational wave detector LIGO (Laser Interferometer Gravitational-Wave Observatory) in 2015 marked the dawn of a new age, in which mankind can directly hear the sounds of the cosmos [1]. Since the ground gravitational wave detection is affected by arm length and Earth vibration, the frequency range of detectable signals is mostly above 10 Hz. However, the 0.1 mHz to 0.1 Hz band contains more astronomical information, such as the coalescence of two supermassive black holes, binary star systems within the Milky Way, and extreme mass ratio spins in extragalactic galaxies [2,3]. Under such a background, the idea of space gravitational wave detection was first proposed by the European Space Agency (ESA) and the National Aeronautics and Space Administration (NASA) [4]. Then, the famous space gravitational wave detection mission LISA (Laser Interferometer Space Antenna) was put on the agenda to establish a million-kilometer interferometer in a relatively “quiet” space environment around 2030, achieving the detection of low-frequency gravitational waves [5,6]. Even more thrilling is the 2015 launch of LISA Pathfinder, a technology verification satellite in the early stages of the LISA mission, which was successfully accomplished [7]. A number of significant scientific findings have also been announced [8,9,10]. This laid the foundation for future related tasks such as ASTROD [11,12], DECIGO [13,14], and BBO [15,16].

At the same time, China’s space gravitational wave detection missions Taiji and TianQin are also gradually taking shape, carrying out research on space gravitational wave detection in different ways [17,18], as shown in Figure 1.

The Taiji program consists of three satellites arranged in an equilateral triangle; the distance between the satellites is 0.3 million km. The satellites adopt a heliocentric orbit, about 20° behind Earth, and the center of mass of the constellation is located in the Earth’s orbit [19,20]. The Taiji program is being implemented in three stages, with the entire constellation set to launch in 2033 as the final goal. Finding and studying gravitational waves in the middle- and low-frequency regions is the ultimate objective [21,22]. After years of research, the first phase of the mission was completed, and the experimental satellite was successfully launched in 2019 at a very low cost and in a short amount of time, successfully verifying key technologies such as the inertial sensor, laser interferometer, and drag-free satellite platform [23,24,25].

The most important technical issue for the space gravitational wave detection program is measurement sensitivity, which is the most important performance requirement for the gravitational reference sensor (GRS) and directly connected to the accuracy that can be attained by the test mass (TM) in free-fall motion. So, it is essential to verify the known and other unknown stray force noise models with high precision in order to meet the designed sensitivity requirements of the Taiji program [26]. Promptly fixing pertinent issues will assist in avoiding a number of risks that are not carried out as envisaged and ensure that the system can conduct typical space experiments. Therefore, it is necessary to construct a test system on the ground to evaluate its performance. This system should be able to mimic the flight environment as closely as possible, which serves not only to confirm the sensitivity level of the design phase but also to forecast the maximum noise level in space, so that it can be closer to the project requirements.

Since its scientific application by Cullen [27], Cavendish [28,29], and Eötvös [30], the torsion pendulum, a classical instrument for measuring weak forces, has been widely used in various fields of precision measurement [31,32] and has been continuously developed and improved. Due to the torsion pendulum’s unique structural properties, it can overcome the influence of Earth’s gravity in the vertical direction while achieving quasi-free fall in the horizontal direction. It has emerged as the first acknowledged option for an international GRS test platform. The University of Trento (Italy), Huazhong University of Science and Technology (China), the University of Florida (USA), and other research institutions have built their own systems to conduct corresponding research.

Among them, the University of Trento [33,34,35] and Huazhong University of Science and Technology [36,37] use a commercial autocollimator for measurement, and their torque measurement resolution can reach 0.1 × 10^−15^ Nm/Hz^1/2^ and 20 × 10^−15^ Nm/Hz^1/2^@1 mHz, respectively. The University of Florida [38,39] adopted the optical lever system measurement method based on PD sensors, the measurement resolution of which can reach 2 × 10^−12^ m/s^2^/Hz^1/2^@0.5 mHz (acceleration). Thus, it can be demonstrated that the optical readout is a highly accurate approach for obtaining the torsion pendulum’s angle information.

In this article, we first provide a thorough introduction to the first stage of the Imech torsion pendulum device’s design, anticipate the system background noise level based on the physical model, and explain the importance of enhancing the readout accuracy for the system’s sensitivity. Second, we developed the interference bench’s hardware structure based on the differential wavefront sensing (DWS) signal to increase the readout accuracy. In combination with the TM, the experiment to measure the background noise of the interferometer was performed to confirm the viability of using the DWS signal to carry out higher-precision GRS noise model verification. Finally, we summarize the effects of the above experimental results on improving the sensitivity of the hybrid system and formulate the future upgrade route based on the apparatus. 

## 2. Torsion Pendulum Design

As of right now, we are aware that the interaction between the Taiji gravitational wave detection program’s GRS and TM is incredibly weak, typically at the fN level, or 16 orders of magnitude less gravity. This calls for the ground measurement to be sufficiently sensitive and able to overcome the effects of the gravitational field.

The torsion pendulum provides an effective method for ground testing of GRS technology [40,41]; the extremely thin suspended fibers with a low torsional spring constant that support the weight of the inertial element hang naturally parallel to the local gravitational field, reducing the coupling of the rotational degrees of freedom of the inertial element with the Earth’s gravitational potential. This makes the transverse bar of the rotational freedom of the suspension fiber axis immune to many external disturbances, especially the pure horizontal or vertical residual airflow, vibration, and electromagnetic effects. It essentially simulates the state of rotational degrees of freedom approaching free fall and allows for the measurement of very small forces acting on the pendulum. At the same time, with the correct selection of fiber sizes and materials, very low torsional spring constants and high quality factors can be achieved, thereby reducing the torque and thermal noise and increasing the sensitivity. In addition, the pendulum helps filter out a portion of the seismic noise, reducing its impact on the measurement.

### 2.1. The Basic Principle of the Torsion Pendulum

Due to the low torsional stiffness of the suspension wire in the vertical direction (Z-axis), the inertial element of the torsion pendulum is insensitive to all degrees of freedom except the rotational motion around Z. Moreover, the resonant frequency of the inertial element is much higher than the measurement frequency band (mHz), so it can be regarded as a rigid body. Figure 2a illustrates its fundamental features. The equation for the test mass’s motion around Z can thus be expressed as follows:(1)$$I\Phi \prime \prime \left(t\right)+\xi \Phi \prime \left(t\right)+\Gamma \Phi \left(t\right)=\mathrm{Torq}\left(t\right)$$
where I stands for the moment of inertia of the TM, ξ is the energy dissipation coefficient (torsion pendulum with an internal damping model; usually equal to 1/Q of the wire material), Γ is the torsional stiffness of the wire, and Φ(t) and Torq(t) are the rotation angle and the torque exerted by the external force acting on the TM, respectively.

The wire’s torsional stiffness Γ is determined by the physical properties of the wire; the expression is shown below: (2)$$\Gamma =\frac{{\pi r}^{4}}{2L}(G+\frac{\mathrm{mg}}{{\pi r}^{2}})$$
where L is the length of the wire, G is the material’s shear modulus, m is the mass of the TM, and g is the value of the gravitational acceleration.

Converting Formula (1) to the frequency domain by Fourier transform, the relationship between the torque and angle can be obtained:(3)$$T(\omega )=(\Gamma \left(1+i\xi \right){+I\omega }^{2})\cdot \emptyset \left(\omega \right)$$

Consequently, the transfer function between them is
(4)$$H\left(\omega \right)=\Gamma \left(1\;-\;{\left(\frac{\omega }{\omega_{0}}\right)}^{2}+\frac{i}{Q}\right)$$
where $\omega_{0}=\sqrt{I/\Gamma }$ is the wire’s natural resonant angular frequency. The following can be seen from Figure 2b: on the left, due to the frequency, H(ω) goes from stable to declining, mainly influenced by $\omega_{0}$; on the right, with the increase in frequency ω, H(ω) increases significantly. This means that a better readout accuracy is needed at a higher frequency to attain the same torque resolution.

Accordingly, any external force acting on the angular frequency ω will be transferred to the scale factor provided by the transfer function described above at the same frequency in the angular motion of the pendulum. Therefore, it is easy to calculate the external torque that excites the pendulum by knowing the parameters of the pendulum and measuring its motion.

### 2.2. Initial Structural Design and Performance Evaluation

The sensitivity of the torsion pendulum is usually expressed in terms of background noise, which is affected by two factors: one is the thermal noise determined by the physical properties of the equipment, and the other is the readout noise of the angle measurement system. 

Then, the background noise amplitude spectrum density (ASD)${\;A}_{\mathrm{Torq}}\left(\omega \right)$ of the torsion pendulum can be expressed in the following form:(5)$$A_{\mathrm{Torq}}(\omega )=\sqrt{S_{\mathrm{Thermal}}{(\omega )+S}_{\phi }\left(\omega \right)\cdot |H(\omega )|}$$
where $S_{\mathrm{Thermal}}\left(\omega \right)$ is the thermal noise power spectral density, and $S_{\phi }\left(\omega \right)$ is the readout noise power spectral density (PSD). Therefore, to improve the sensitivity of the torsion pendulum, it is necessary to take measures with the two aims of reducing thermal noise and improving the readout accuracy.

For thermal noise, the torque generated by it is entirely from the suspension wire dissipation of the system itself [42], which is unavoidable and is the main factor affecting the improvement of the torsion pendulum’s sensitivity. Its basic physical description is as follows:(6)$${\;S}_{T_{\mathrm{th}}}\left(\omega \right){=4K}_{B}T\frac{\Gamma }{\omega Q}$$

K_B_ = 1.38 × 10^−23^(J/K) is the Boltzmann constant, and T is the operating temperature. 

On the basis of the theoretical foundation mentioned above, we characterized the torsion pendulum’s performance level under the influence of thermal noise and autocollimator readout noise, as shown in Figure 3.

According to the curve above, readout noise has a much smaller impact than thermal noise in the frequency range below 0.01 Hz, which is where the torque and angle measurements are made (different instruments may have minor variations in this range). The readout noise is the key constraint for the frequency range above 0.01 Hz, and lowering the readout noise level can increase the sensitivity of the torsion pendulum. As a result, efforts should be made to reduce the noise of the torsion pendulum in measurements in this range. 

Thus, it is clear that the mass of the TM, the material, length, and diameter of the wire, and other environmental parameters directly affect the thermal noise of the apparatus, which can be reduced through reasonable design to achieve the purpose of improving the sensitivity, but subject to various experimental conditions this effect is actually very limited.

Based on the above principles, we carried out the basic structural design of the Imech torsion pendulum, as shown in Figure 4. The corresponding design parameters and environmental conditions are shown in Table 1.

The torsion pendulum is primarily made up of a bottom plate, three fixed stems, a mechanism for changing the wire’s state, and a TM as the inertial component of the system. It has a cubic form with a connecting thread hole on the top surface that makes it simple to attach with the suspension wire. To reduce the total mass of the inertial components, aluminum alloy is employed, and high-precision surface quality may be easily attained. The TM is directly utilized in the angle measurement as a reflector and has one surface in the X direction polished to increase laser reflectivity. The structure for adjusting the inclination of the TM is additionally positioned above it to restrain the effect of the simple pendulum motion. 

A high-elastic-modulus, 99.95% pure tungsten wire from the Goodfellow Company was used to make the suspension wire. The quality factor Q can reach 2000 with the appropriate heat treatment. Considering the inertia elements’ mass, the wire material’s strength limits, and the experimental environment, the diameter and length of the tungsten wire were designed to be 50 μm and 0.84 m, respectively, and both ends were connected to the suspension column by clamping and fixing.

For the Imech torsion pendulum, the thermal noise curve can be obtained using the parameters in Table 1, and the readout noise primarily depends on the readout system’s measurement precision. The readout equipment is typically an autocollimator, which follows a dependable and straightforward procedure. Currently, multiple models of the Trioptics autocollimator Triangle can measure resolutions greater than 50 nrad/Hz^1/2^ or 500 nrad/Hz^1/2^. On this premise, the torsion pendulum’s sensitivity can reach 5 × 10^−15^ Nm/Hz^1/2^, particularly for the frequency band on both sides of 6 mHz, as shown in Figure 5. The sensitivity of the torsion pendulum improves dramatically as the readout accuracy improves, so it is necessary to develop a new optical readout device. Base on a long-term heterodyne interferometer as a development foundation, we expand the laser interferometry technology based on the DWS signal in detail to improve the torsion pendulum’s measurement sensitivity in the following section.

## 3. DWS Interferometer Optical Readout System

### 3.1. Design of the Interferometer

In comparison to the traditional Michelson interferometer [43], the laser interferometer based on the DWS signal provides better anti-interference capability and measurement accuracy in the order of pm. In conjunction with the topic in Section 2, the laser interferometer system was constructed, consisting of three components: a laser source, an optical bench, and a data collection module. The optical bench included in it developed the LISA Pathfinder’s pattern to a 3D level [44], which is near to the ultimate design form of the Taiji program and offers a basis for further study.

The Nd:YAG solid-state laser source was a LUMENTUM Company 125/126 series NPRO narrow-linear-width and low-noise solid laser with a wavelength of 1064 nm. As illustrated in Figure 6a, the laser is divided into two halves by the fiber beam splitter, and the frequency difference between the two channels is modulated to 1 kHz by the acousto-optic modulators (AOMs) and sent into the optical bench.

The notion of sharing optical channels is commonly used in the process of optical bench construction to reduce systems’ weight, volume, and number of components. Four interference optical routes were created and labeled as RT1, -2, -3, and -4. RT1 and RT2 were equal-arm interferometers that were used to evaluate the optical bench’s stability, while RT3 and RT4 were unequal-arm interferometers that were used to test phase change and mass displacement change, respectively. To facilitate the combination with the torsion pendulum, the optical path of the TM interferometer was rotated 45 degrees through a right-angled prism in the center and transmitted perpendicular to the interferometer bench until it reached the TM and returned. As seen in Figure 6b, after the PBS, it transformed into S light and interfered with the nearby laser. The interferometer also consisted of the TM, which was the terminal optical component of RT3. To precisely measure the TM’s rotation angle for the torsion pendulum experiment, we only needed to use the RT1 and RT3 interferometers. The interferometer correlation evaluation test could be performed using the remaining two channels.

In terms of data acquisition, the four-channel laser signal interferes in space, and the quadrant photoelectric detector (QPD) detects and reads the phase and amplitude changes of the interference signal at the end of the optical path and then calculates the angle changes of the TM. Eight QPDs were installed in the terminal to back one another up and increase the system’s reliability for the consideration of engineering applications. In order to address the tiny angle shift brought on by the test mass, the QPD determined the phase change of the interference signal brought on by the motion of the TM and communicated the voltage signal to the self-developed phase meter. The phase meter’s phase resolution precision can reach 0.5 rad/Hz^1/2^ [45], which can satisfy the needs of the present measurements.

### 3.2. Resolution Test of the Laser Interferometer Measurement System

To reduce the influence of environmental thermal stability on the system, a pair of laser interferometer prototypes were developed using an invar substrate and a few fused quartz optical elements [46]. By using a coordinate measuring machine (CMM) and the calibrated quadrant photodiode pair (CQP) to achieve accurate alignment, the final angle error and position error reached 100 μrad and 50 μm, respectively [47]. One interferometer twin was successfully applied to Taiji-1 and produced good in-orbit test results [48]. We then employed the other one for ground testing to evaluate the angle resolution performance of its own system.

The vacuum test system at the HuaiRou Gravitational Wave Experimental Center, Institute of Mechanics, Chinese Academy of Sciences, served as the experimental facility for the interferometer resolution test. As shown in Figure 7 and Figure 8, the system consisted of three 3m-diameter interconnected tanks set on a base to isolate vibrations. For our experiment, we used only one of these tanks. The apparatus can produce a vacuum of 10^−4^ Pa. To isolate high-frequency ground vibration, an air-floating platform was placed inside the tank. The optical bench and its adjustment mechanism were mounted as a single unit on the vibration isolation platform, while the TM as a reflector was fixed on the same platform by means of its support structure. The laser was deflected through the prism to the TM and returned along the original path; this alignment process was realized by the optical bench’s adjustment mechanism. Finally, the laser and electrical signals were routed via the silo flange to the system.

To limit the impact of environmental stability on the system even further, the test experiment was conducted on weekends and evenings in a silent vacuum atmosphere, and all of the pumps of the system had been shut down before the experiment. The experiment lasted for several nights, and we finally took two random pieces of data to illustrate the results. The phase meter solved the voltage signal and obtained the background noise in the time domain of the associated angle measurement, as illustrated in Figure 9. In addition, we performed a simple data smoothing on the basis of the original data to further remove the system and environmental noise effects contained in the phase measurement. It can be seen that the stability of angle measurement of the heterodyne interferometric optical readout system based on the DWS signal can reach μrad level without any noise compensation mechanism.

To make the resolution of the system at various frequencies clear, phase noise was translated from the time domain to the frequency domain by fast Fourier transform (FFT), as shown in Figure 10. The angle measurement accuracy of the laser interferometer based on the DWS signal exceeds 10 nrad/Hz^1/2^ in the frequency band of 0.1 mHz–1 Hz and can reach 4 nrad/Hz^1/2^@10 mHz, which is better than commercial autocollimator products. As a result, a larger variety of frequency bands can be used to implement the background noise calibration of the torsion pendulum, and a novel method can be offered for greater accuracy measurement.

## 4. Conclusions and Outlook

In this study, we proposed a new weak force measurement system based on DWS optical readout that consists of a torsional pendulum and a laser interferometer. The thermal noise limit is one of the important elements impacting the sensitivity of the torsion pendulum, which cannot be surpassed due to physical constraints and can only be enhanced to a limited extent through rational design.

At the same time, the influence of different precision autocollimators on the sensitivity of the Imech torsion pendulum was compared successively. It improved by almost one order of magnitude, most notably in the high-frequency band, while the angle readout noise increased by one order of magnitude, except for the resonant frequency. As a result, we concentrated on enhancing the angle readout accuracy to reduce the system’s readout noise. Then, we constructed a laser interference system based on heterodyne interference technology and incorporated it into the torsion pendulum system to enhance the angle readout accuracy.

The interferometer bench was added to finally form a prototype that could be applied in engineering. We performed a background noise calibration experiment in conjunction with the TM, and the findings demonstrated that the interferometer’s accuracy was better than 10 nrad/Hz^−1/2^ across the whole frequency range. As illustrated in Figure 11, the equivalent torque sensitivity of the torsion pendulum system can approach 5 × 10^−15^ Nm/Hz^1/2^@10 mHz when combined with the thermal noise. The measurement accuracy for low-frequency bands was found to have essentially little difference under the influence of the transfer function; however, the improvement effect was evident for the high-frequency bands, which still makes a lot of sense for the torsion pendulum measurement.

It can be seen that the performance of the optical readout system based on the DWS signal is superior to the current conventional angle measurement methods. This will also help us to optimize and upgrade the system, carry out subsequent research on predicting the upper limit of stray force noise, and provide more convenient conditions for guiding the development of future Taiji program-related issues in space. The fact that the Earth’s gravitational field cannot be shielded causes this type of experiment to be no less difficult to conduct on the ground than in space.

Similarly, the sensitivity of the torsion pendulum as a GRS test instrument is determined by the background noise spectrum of the GRS in a free state. For a space gravitational wave detection GRS, the acceleration limit noise generated by the background can be determined by dividing the rotation radius of the inertial mass on this basis.

In the future, we will use a high-accuracy optical readout system of DWS signals to measure the effects of different external disturbances acting on the TM. Currently, the system is being upgraded in many respects, such as shock absorption, heat insulation, temperature control, and electromagnetic shielding, allowing us to measure weak forces with greater accuracy. This can meet the needs of future space gravitational wave detection, gravitational field measurement, and other related research content.

## Figures and Tables

**Figure 1 sensors-23-08087-f001:**
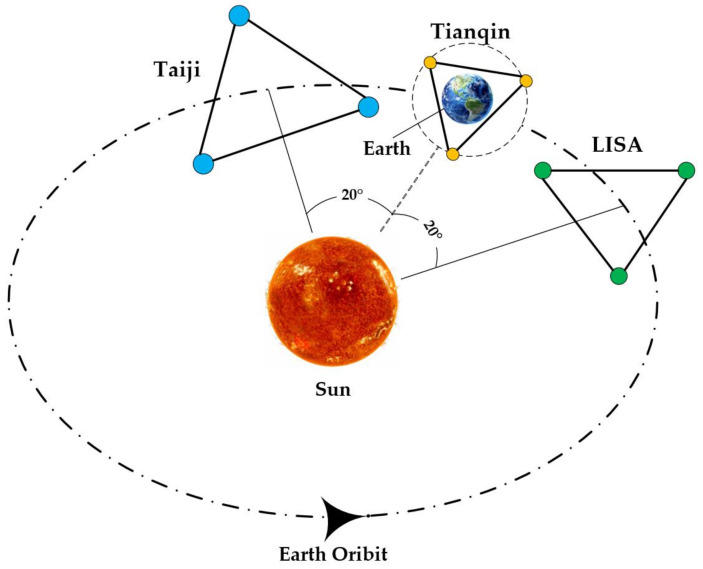
An orbital arrangement diagram for the LISA, Taiji, and TianQin missions. All of their satellite formations are equilateral triangles, with TianQin sharing the Earth’s orbit and LISA and Taiji following the solar orbit, 20° in front of and behind the Earth, respectively.

**Figure 2 sensors-23-08087-f002:**
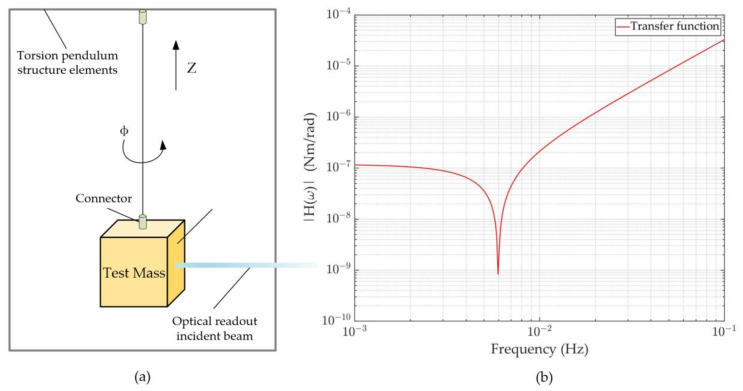
The basic principle of the torsion pendulum and its transfer function: (**a**) The torsion pendulum is equipped in the vacuum chamber, using the optical readout directly to gain the angle information. (**b**) Transfer function between torque and angle; smaller values mean lower torque can be distinguished.

**Figure 3 sensors-23-08087-f003:**
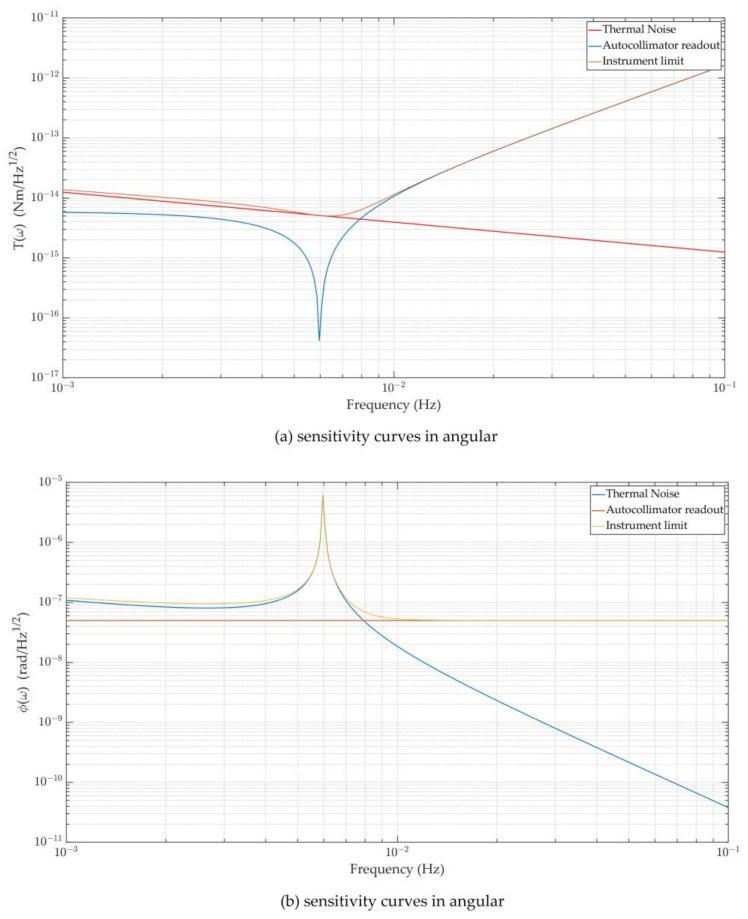
The instrument limits of the torsion pendulum, as synthesized from thermal noise and optical readout noise: (**a**) system performance in the form of torque measurement; (**b**) system performance in the form of angular measurement.

**Figure 4 sensors-23-08087-f004:**
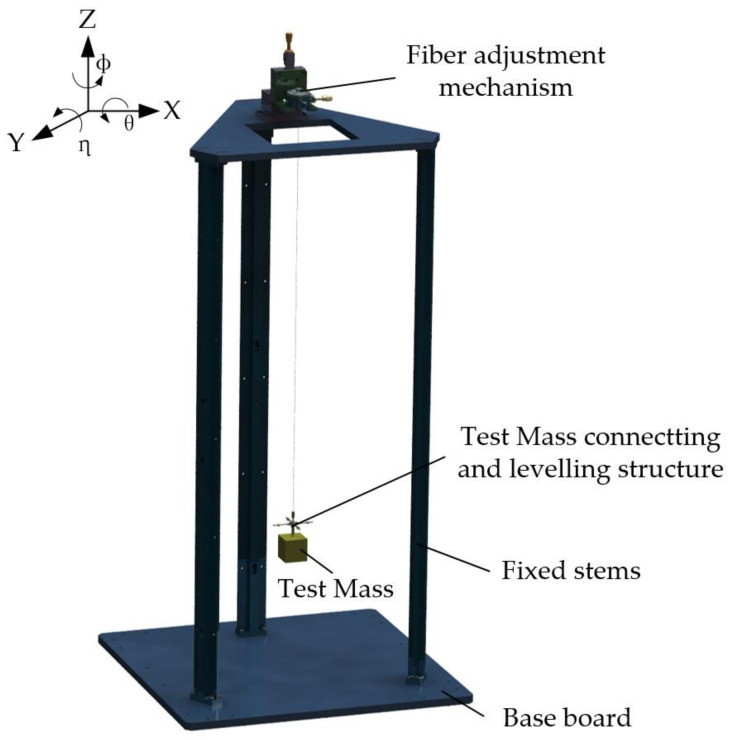
Basic structure of the Imech torsion pendulum: The TM is suspended from a rigid support by a fine wire, and its position can be adjusted by the mechanism provided. The mechanism has X/Y/Z translation capabilities and rotating motion about the Z-axis.

**Figure 5 sensors-23-08087-f005:**
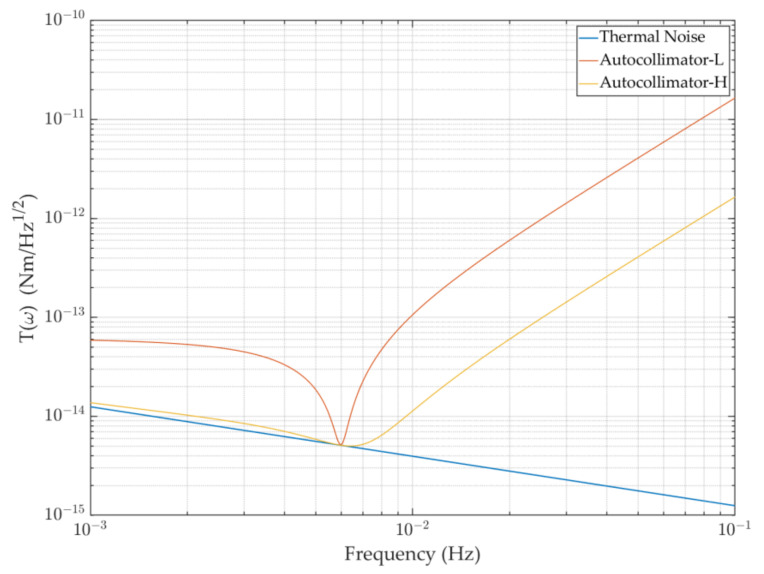
Background noise diagram of the Imech torsion pendulum: The blue curve is the thermal noise ASD of the apparatus, the brown curve is the apparatus’ background noise under the low-precision autocollimator readout condition, and the yellow curve is the background noise under the high-precision readout condition.

**Figure 6 sensors-23-08087-f006:**
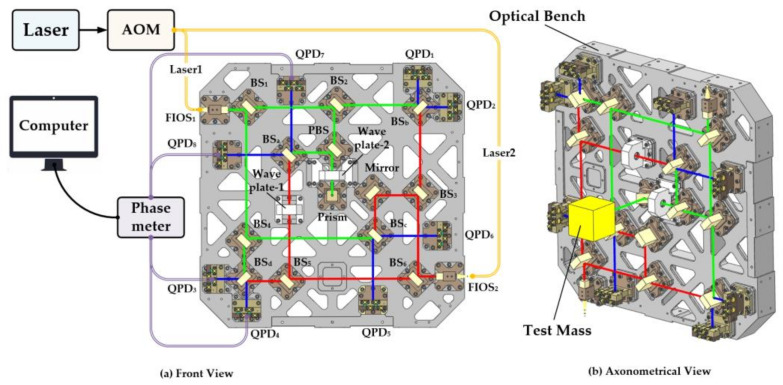
Laser interferometer bench design diagram; the laser routes are as follows: RT1: FIOS_1_ (fiber injector optical subassembly)-BS_1_ (beam splitter)-BS_2_-BS_b_-QPD_1/2_ and FIOS_2_-BS_6_-BS_3_-BS_b_-QPD_1/2_; RT2: FIOS_1_-BS_1_-BS_4_-BS_d_-QPD_3/4_ and FIOS_2_-BS_6_-BS_5_-BS_d_-QPD_3/4_; RT3: FIOS_1_-BS_1_-BS_2_-PBS (polarized beam splitter)-WP_2_-prism-TM-prism-WP_2_-PBS-BS_a_-QPD_7/8_ and FIOS_2_-BS_6_-BS_5_-WP_1_-BS_a_-QPD_7/8_; RT4: FIOS_1_-BS_1_-BS_4_-BS_d_-QPD_3/4_ and FIOS_2_-BS_6_-BS_5_-BS_d_-QPD_3/4_; RT3: FIOS_1_-BS_1_-BS_4_-QPD_5/6_ and FIOS_2_-BS_6_-BS_3_-mirror-BS_c_-QPD_5/6_. Green lines are the lasers emitted by FIOS_1_, red lines are the lasers emitted by FIOS_1_, and blue lines are the interference lights.

**Figure 7 sensors-23-08087-f007:**
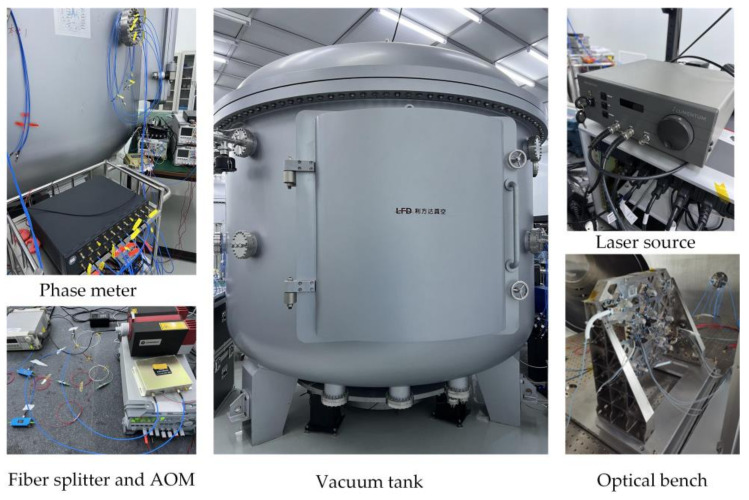
Experimental devices of the laser interferometer: the middle is one of the tanks of the vacuum system, the upper left is the self-developed phase meter, the lower left is the AOM developed by CETC and the fiber splitter purchased from Thorlabs, the upper right is the ultra-stable laser, and the lower right is the interferometer bench.

**Figure 8 sensors-23-08087-f008:**
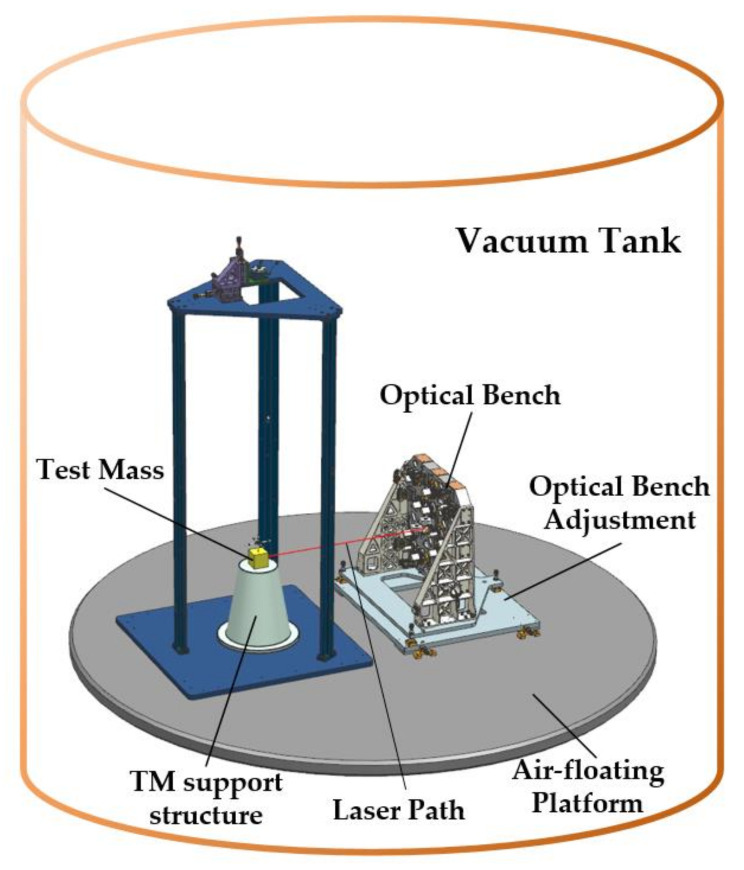
Layout of equipment elements inside the vacuum tank: the optical bench achieves the alignment to the TM by means of the adjustment mechanism, and the red line indicates the laser path constructed between them.

**Figure 9 sensors-23-08087-f009:**
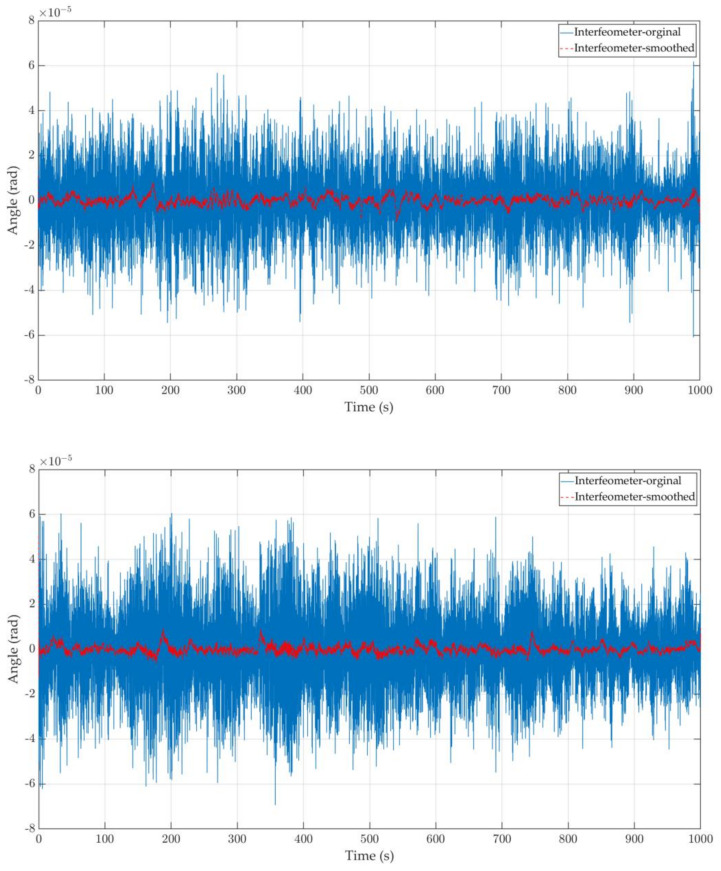
Background noise curve measured by the interferometer angle in the time domain in two different time periods.

**Figure 10 sensors-23-08087-f010:**
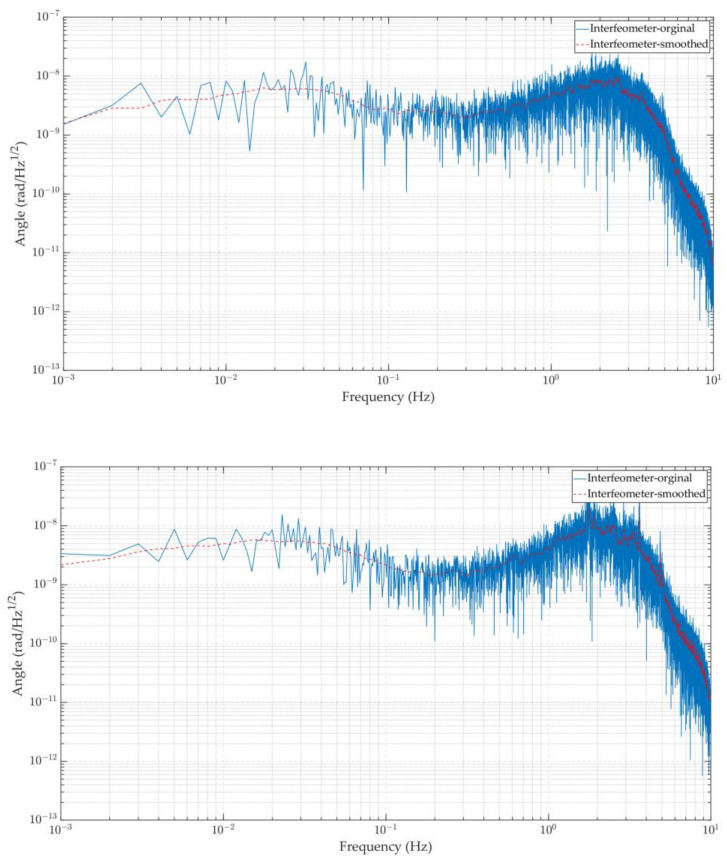
Background noise curve measured by the interferometer angle in the frequency domain in two different time periods.

**Figure 11 sensors-23-08087-f011:**
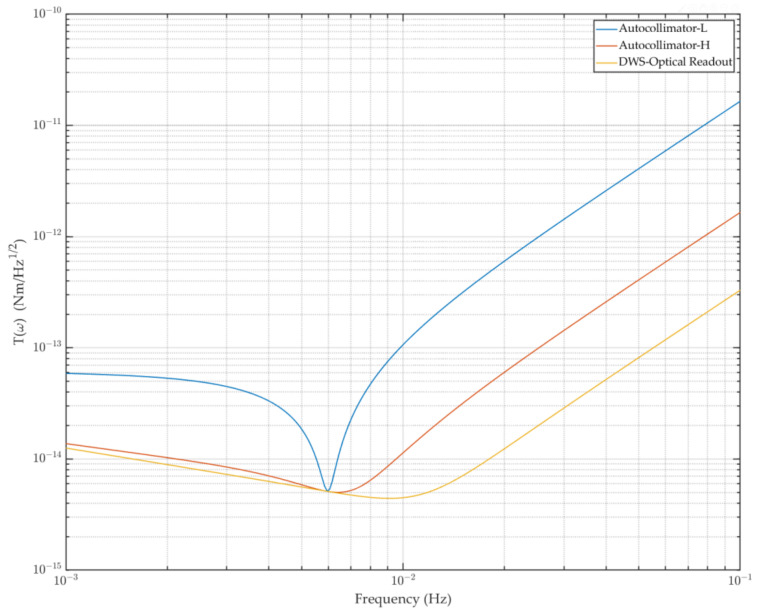
Background noise curves of the torsion pendulum under different readout conditions, synthesized with the thermal noise. The blue curve is the apparatus’ background noise under the low-precision autocollimator readout condition, the orange curve is the noise under the high-precision autocollimator readout, and the yellow curve is the noise under the DWS optical readout condition.

**Table 1 sensors-23-08087-t001:** Basic parameters of the Imech torsion pendulum.

	Parameters	Symbol	Values	Units
Test mass	Length of sides	d	46	mm
	Mass	m_total_	0.238	kg
Tungsten wire	Length	L	0.841	m
	Shear elasticity	G	161	GPa
	Radius	r	25	μm
	Quality factor	Q	2000	/
Environment conditions	Gravitational acceleration	g	9.801	m/s^2^
	Temperature	T	300	K

## Data Availability

Not applicable.

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
