# Peer review of "Using DWS Optical Readout to Improve the Sensitivity of Torsion Pendulum"

_sensors, 2023, doi:10.3390/s23198087_

Round 1
Reviewer 1 Report
This paper described using the DWS to improve the readout noise of a torsion balance in view of applying the technique for the space GW detectors. The experiments showed an improvement in noise reduction compared with the commercial autocollimator readout. It would be or interest to the community. However, it is not clearly stated if this system is something similar to the LISA system or if the scheme was first proposed here for such application.
It would be helpful to give a comparison with LISA’s readout noise floor.
Some comments embedded with the pdf file is attached.

The English will need to be improved. I only marked a few places
Reviewer 2 Report
This manuscript reported the technique for enhancing the sensitivity of the torsion pendulum system by employing the differential wavefront sensing (DWS) optical readout and the comparative experiments were conducted. My review comments are as follows:
1. In order to demonstrate the effectiveness of the technology used, multiple comparative studies and experiments were conducted, and the experimental results are shown in Figures 3, 5, and 10. However, there is a lack of description on how the comparison curve in the figure were obtained, which needs to be supplemented.
2.Figure 6 shows the experimental system in two views, but the two views seem to lack a reasonable correspondence and cannot clearly display the mechanical structure and optical path. Need to be redesigned according to standards.
Minor editing of English language required
Round 2
Reviewer 2 Report
The paper has been properly revised and it is recommended to accept.